# Full-Length Transcriptome Analysis of Alternative Splicing and Polyadenylation in the Molecular Regulation of Labor Division in *Apis cerana cerana*

**DOI:** 10.3390/ijms26167859

**Published:** 2025-08-14

**Authors:** Dan Yao, Yuanchan Fan, Wencai Zhou, Hongpin Zhan, Yinglong Yu, Xiaoping Wei

**Affiliations:** 1Guizhou Institute of Integrated Agriculture Development, Guizhou Academy of Agricultural Sciences, Guiyang 550006, China; dyao20241218@126.com (D.Y.); zhan_hp@126.com (H.Z.); yuyinglong2012@163.com (Y.Y.); 2Apicultural Research Institute, College of Animal Science and Technology, Yangzhou University, Yangzhou 225009, China; yuanchanfan1997@163.com

**Keywords:** *Apis cerana cerana*, alternative splicing, alternative polyadenylation, development, functional transformation, division of labor

## Abstract

Honeybees are vital pollinators with functional differentiation as a key survival strategy. The Chinese honeybee (*Apis cerana cerana*) exhibits exceptional nectar foraging in complex terrains, yet how alternative splicing (AS) and polyadenylation (APA) regulate its labor division remains unclear. Here, we applied PacBio full-length transcriptome sequencing to annotate worker bee transcriptomes across three developmental stages (*Ac*3d, *Ac*10d, *Ac*21d), calibrating the third-generation sequencing data with second-generation sequencing to enhance transcriptome annotation accuracy. We identified 17,961 isoforms and 1922 APA genes, finding that alternative first exon was the major type of AS, while APA enhances transcriptomic diversity via dual polyadenylation sites in most genes. Functional analyses revealed AS enrichment in growth signaling (*Vg6, CYP15A1*) and immune pathways (*PTPRR*), whereas APA regulated growth signaling (*Vg6*), energy metabolism (*Acsl1, AcceFE*), and oxidative stress (*PTPRR, PPO2*). Validation by PCR and 3′RACE confirmed stage-specific AS/APA events in key genes. These findings significantly enhance the *A. cerana cerana* reference genome annotation and provide valuable insights into the mechanisms of AS and APA regulation underlying honeybee development and functional transitions.

## 1. Introduction

Honey bees are eusocial insects with a high degree of division of labor. The life cycle of worker bees involves different functional stages. Young workers, known as nurse bees, primarily engage in brood rearing. As workers develop, they transition to foraging activities, becoming forager bees, which exhibit division of labor for both pollen and nectar collection [1]. The functional shift of adult worker bees from behaviors within the nest to outside of the nest is accompanied by changes in brain structure and neural function, alterations in hormone levels, and changes in protein and gene expression [2,3,4,5]. Previous studies have indicated that division of labor within a colony is regulated by complex, multifaceted factors, involving many important regulatory processes that are precisely controlled by genes in the honeybee brain or other tissues. In terms of environmental factors, colony strength, intra-colony pheromones, environmental temperature, and pathogen invasion can regulate the functional transition of honeybees [6,7]. For instance, variation in the proportion of forager bees, number of young honeybees, and food storage levels within a colony, especially during critical periods of brood rearing and nest building, can lead to changes in division of labor [8,9,10]. Brood pheromones can regulate the balance of nectar and pollen collection by forager bees [11,12].

Behavioral maturation is accompanied by many changes in physiology and neural function, including body weight, flight metabolism, sugar response capability, hormone levels, brain chemicals and structure, and circadian rhythms. Withers et al. [13] found that the number of mushroom body Kenyon cells in the brains of Italian bee (*Apis mellifera ligustica*) foragers is significantly lower than that in nurse bees, and the number of neural fiber networks is significantly higher. This indicates a relationship between the structural changes in the mushroom body and division of labor. Sullivan et al. [14] found that juvenile hormone (JH) in honeybees can promote the transition to foraging behavior, suggesting that JH may act directly on the nervous system or influence multiple physiological systems to affect the maturation of honeybee behavior. Division of labor in honeybees is also related to biogenic amines, with higher concentrations of dopamine, octopamine, and 5-hydroxytryptamine detected in foragers than in nurse bees [15,16].

Furthermore, the functional transformation of honeybees is regulated by multiple genes. Previous studies have found that animal circadian rhythm genes, such as Period (*Per*), acetylcholinesterase (*AChE*), cGMP-dependent protein kinase (*cGMP*), Foraging (*For*), Thioester-containing protein II (*Tep2*), Rab protein 7 (*Rab7*), possible cytochrome 4506g2 (*p4506g2*), larval series protein 2 (*lsp2*), Histone H3.3A and Histone3.3B (*His3.3A* and *His3.3B*), synaptotagmin (*syt1*), and the microRNAs (miRNAs) *ame-mir-276*, *ame-mir-2796*, *microrna-932*, and *ame-mir-279* regulate the functional transition of *Apis mellifera* by regulating the JAK/STAT pathway [17].

*Apis cerana cerana*, a subspecies adapted to diverse terrestrial ecosystems, exhibits age-based labor division similar to *Apis mellifera* but with unique foraging strategies in complex terrains. It is an excellent pollinator with significant ecological, economic, and research value. There are differences in the division of labor among worker bees of different ages in *A. cerana cerana*; orientation flights at 3–5 days of age and foraging for nectar and pollen begin at 10 days old, with the strongest foraging ability around 21 days old [18]. Extensive research has focused on the growth, development, and cognitive behavior of the Chinese honeybee, yielding a series of important advances. While functional transitions in *A. cerana cerana* have been partially described, the post-transcriptional regulatory mechanism particularly alternative splicing (AS) and polyadenylation (APA), underlying this process remain entirely uncharacterized.

A large number of recent studies have demonstrated the important role of post-transcriptional regulation in insect growth, development, and behavior. AS and APA are two important mechanisms of post-transcriptional regulation. AS refers to the process by which a gene produces multiple different mRNA transcripts, thus encoding different proteins. This phenomenon exists widely in eukaryotes and is an important mechanism to increase protein diversity and regulate gene expression. There are seven types of alternative splicing events: skipped exon (SE), mutually exclusive exon (MX), alternative 5′ splice site (A5), alternative 3′ splice site (A3), alternative first exon (AF), alternative last exon (AL), and retained intron (RI). This process plays an important role in development in insects. The Notch signaling pathway has multiple functions in insect development, including wing and leg development. Notch in the brown planthopper is a complex alternatively spliced gene that has at least 3 transcription start sites, 4 exon skips, and 21 transcription end sites. It uses these sites to form variants and encode a series of proteins, thereby regulating the formation of wings and legs [19]. APA refers to the process by which the 3′ end of different transcripts from the same gene can vary due to AS and APA, thereby increasing the diversity of plant and animal transcriptomes. In eukaryotes, polyadenylation causes the termination of mRNA molecules at the 3′ end. The poly(A) tail protects mRNA from exonuclease attack and is essential for transcription termination, mRNA export from the nucleus, and translation. APA regulates gene expression levels and stability by producing mRNA variants with different 3′ ends, which is crucial for the regulation of gene expression during cellular differentiation, proliferation, and apoptosis.

Third-generation sequencing technologies, represented by PacBio and Nanopore, mark a significant advancement in single-molecule sequencing. PacBio SMRT (Single Molecule Real-Time) technology benefits from long read lengths and the ability to generate reads of thousands to tens of thousands of base pairs. This technology provides comprehensive information about gene expression, transcript structure, and gene regulatory networks and is particularly adept at identifying different isoforms produced by AS, mining APA to discover new genes and transcripts, and precisely locating fusion genes. Yang et al. [20] employed PacBio sequencing to identify 247 differential APA events between the early and adult stages in *Litopenaeus vannamei* and further revealed five dynamic APA patterns. Full-length transcriptome sequencing, owing to its ultra-long read lengths and the lack of assembly steps, can yield complete transcript information, providing a powerful tool for in-depth analyses of the complexity and dynamic changes in gene expression.

To address the lack of research on the molecular regulatory mechanisms underlying adult Chinese bee development and functional transformation, this study utilized PacBio sequencing technology to sequence the transcriptomes of individuals at three representative age points during the functional transformation of *A. cerana cerana*. Transcriptome complexity was evaluated, including AS and APA. We annotated differential AS genes (DASs) and differential APA genes (DAPAs), compared highly expressed AS and APA regulatory genes using second-generation transcript data, and evaluated pathways and genes related to development and functional transformation in *A. cerana cerana*. Finally, key regulatory AS and APA genes were verified through PCR and 3′RACE. The study results will aid future research aimed at elucidating the mechanisms underlying adult bee development and functional transformation.

## 2. Results

### 2.1. AS Functional Annotation

In the *Ac*3d, *Ac*10d, and *Ac*21d groups, we obtained 4695, 6417, and 6849 alternative splice isoforms from 2556, 2732, and 2987 genes. The GO analysis of these AS genes revealed 55, 54, and 56 terms in the *Ac*3d (Figure 1A), Ac10d (Figure 1B), and Ac21d (Figure 1C) groups. These AS genes were significantly enriched in the following terms: binding (833, 764, 932), cellular process (706, 654, 793), metabolic process (638, 594, 692), catalytic activity (586, 540, 652), single-organism process (528, 506, 600), and membrane (403, 372, 443). In addition, the KEGG pathway analysis indicated that AS genes were involved in 137 (Figure 1D), 137 (Figure 1E), and 137 (Figure 1F) pathways. In the early development stage of *A. cerana cerana*, especially at *Ac*3d and *Ac*10d after hatching, the main enriched pathways were identified as energy metabolism pathways, in which glycophospholipid metabolism plays a central role. The difference is that on *Ac*21d, the main enrichment pathway changes to a biological process, which was mainly characterized by ubiquitin-mediated protein degradation.

### 2.2. Differential AS Events

The full-length transcriptome data for *Ac*3d, Ac10d, and Ac21d revealed seven types of AS: 1523 SE, 162 MX, 1548 A5, 1625 A3, 932 RI, 1626 AF, and 208 AL (Figure 2A). The pairwise comparisons between *Ac*3d, *Ac*10d, and *Ac*21d showed that splicing events were enriched in AF, suggesting that AS is involved in the regulation of the first exon during bee growth, development, and functional transformation. AS is highly spatiotemporally specific. Accordingly, we quantified differential splicing events by pairwise comparisons of individual tissues at different developmental ages in combination with Illumina sequencing data. A total of 20, 19, and 24 DASEs were identified in *Ac*3d vs. *Ac*10d, *Ac*10d vs. *Ac*21d, and *Ac*3d vs. *Ac*21d comparison groups, respectively (Figure 2B). Additionally, Venn analysis showed that there was 1 DASE in the aforementioned three comparison groups, whereas the numbers of unique DASEs were 6, 13, and 13, respectively (Figure 2B).

### 2.3. Key DASEs Related to Development

DASEs related to growth, development, and functional transformation were further evaluated. DASEs for three genes were identified, including vitellogenin 6 (*Vg6*) (gene id: LOC107998857), tyrosine protein phosphatase (*PTPRR*) (gene id: LOC107998988), and methyl farnesoate epoxidase (*CYP15A1*) (gene id: LOC107999579) (Figure 3). With the shift in workers from nest work to collection work, these genes related to development were gradually inhibited. DASEs were verified by RT-PCR (Figure 4).

### 2.4. Profiling of Alternative Polyadenylation Sites

Polyadenylation sites on 1922 genes were analyzed, revealing distinct distributions of APA sites among treatment groups (Figure 5A). Further analysis indicated that the same gene exhibited variation in the number of APA sites among the three sample groups. Five motifs, AAUAAA (Figure 5B), GUAC (Figure 5C), UUUUUUUU (Figure 5D), UGUAA (Figure 5E), and AAUAAA (Figure 5F), were identified in the region upstream of the APA sites in the three sets of full-length transcript data for *A. cerana cerana*. A total of 21, 20, and 30 DAPAGs were identified in *Ac*3d vs. *Ac*10d, *Ac*10d vs. *Ac*21d, and *Ac*3d vs. *Ac*21d comparison groups, respectively (Figure 5G). Additionally, Venn analysis showed that there was 1 DAPAG in the aforementioned three comparison groups, whereas the numbers of unique DAPAGs were 7, 17, and 17, respectively (Figure 5G).

### 2.5. APA Functional Annotation

The gene annotation results using the GO database showed that APA genes were highly significantly enriched in 56, 54, and 56 terms at *Ac*3d (Figure 6A), *Ac*10d (Figure 6B), and *Ac*21d (Figure 6C). Specifically, these genes were notably enriched in the following biological functions: binding (865, 657, 810), cellular processes (731, 555, 671), metabolic processes (674, 538, 629), catalytic activities (634, 504, 583), single-organism processes (578, 464, 523), and membrane (425, 331, 391). According to KEGG pathway annotation results, APA genes were associated with a total of 131 (Figure 6D), 126 (Figure 6E), and 132 (Figure 6F) pathways, including metabolism, organismal systems, environmental information processing, cellular processes, and genetic information processing. In *Ac*3d and *Ac*10d, the main enriched pathways were metabolic pathways. In *Ac*21d, the main enrichment pathway was amino sugar and nucleotide sugar metabolism.

### 2.6. Key DAPAGs Related to Development

APA regulates the expression level and stability of genes by generating mRNA variants with different 3′ ends. This was essential for the regulation of gene expression in the processes of cell differentiation, proliferation, and apoptosis. We identified, for the first time, growth and development-related genes in *A. cerana cerana* regulated by APA. In total, 54 DAPAGs were found in *Ac*3d, *Ac*10d, and *Ac*21d, including 5 DAPAGs related to growth and development: long chain fatty acid CoA ligase 1 gene (*Acsl1*) (gene id: LOC108004007), esterase FE4 gene (*AcceFE*) (gene id: LOC107998473), *Vg6* (gene id: LOC107998473), receptor-type tyrosine-protein phosphatase R gene (*PTPRR*) (gene id: LOC107998988), and phenoloxidase 2 gene (*PPO2*) (gene id: LOC108004136) (Figure 7A,B). DAPAGs were verified by 3′ RACE (Figure 7C).

## 3. Discussion

During development, alternative splicing (AS) and alternative polyadenylation (APA) enable individual genes to generate multiple protein variants. Despite widespread use of NGS for honeybee gene expression profiling, understanding of post-transcriptional regulation (e.g., AS, APA) remains limited due to NGS technical constraints such as short reads and incomplete transcript coverage. We employed PacBio third-generation sequencing technology that provides full-length transcript characterization, enabling precise genome-wide identification of AS events and APA genes in *A. c. cerana*. Through re-annotation of the full-length transcriptome across three critical developmental stages of division of labor (*Ac*3d, *Ac*10d, and *Ac*21d), we identified 17,961 distinct isoforms and 1922 APA-regulated genes. The integration of PacBio long-read sequencing with Illumina short-read data establishes a comprehensive framework for investigating how post-transcriptional regulatory mechanisms govern division of labor in honeybees.

Variation in the proportion and types of ASEs was observed across different developmental stages. AS is the process by which an mRNA precursor is spliced at different splice sites to produce diverse mRNA splice isoforms. This is a crucial mechanism for regulating gene expression and generating proteome diversity, and it accounts for the significant discrepancy between the number of genes and proteins in eukaryotic organisms. Extensive research has shown that AS genes are prevalent across a wide range of taxa, including plants, animals, and microorganisms. For instance, 61% of genes in *Arabidopsis thaliana* undergo AS [21]; in humans and pigs, 95% and 30% of genes undergo AS, respectively [22,23]; 45% of the *Drosophila* genome is subject to AS [24]; and five AS genes have been identified in *Nosema ceranae* [25].

Focusing on *A. cerana cerana* at 3, 10, and 21 days old, this study detected 4695, 6417, and 6849 isoforms, respectively, with 2556 (54.44%), 2732 (42.57%), and 2987 (43.61%) genes undergoing AS events. These results suggest that the proportion of genes involved in AS events varies among bees at different developmental stages. Furthermore, the types of AS also vary in frequency among species [26]. In *A. mellifera*, the alternative 3′ splice site is the most common AS type in endocytosis-related genes [27]. In this study, the distribution of AS event types was as follows: 1523 SE (19.98%), 162 MX (2.12%), 1548 A5 (20.30%), 1625 A3 (21.31%), 932 RI (12.22%), 1626 AF (21.33%), 208 AL (2.73%), AF (21.33%), and A3 (21.31%). Additionally, two genes associated with growth and development were randomly selected for verification using RT-PCR. Two transcripts were obtained for both gene *Vg6* and *LOC107998857_novel04*, matching the sizes predicted from sequencing data (Figure 4), confirming the authenticity and reliability of the AS events detected in *A. cerana cerana* genes in this study. Notably, our finding that alternative First Exon (AF) constitutes the most abundant AS type (21.33% of events, Figure 2A) contrasts with a prior *A. cerana* transcriptome study using Nanopore sequencing, which reported Intron Retention (IR) as predominant [28]. This divergence likely reflects both methodological and biological factors. Technically, PacBio SMRT sequencing—coupled with Illumina-based correction—provides superior 5′-end resolution, enhancing AF detection through precise transcriptional start site annotation. Nanopore platforms, while valuable for full-length profiling, exhibit higher error rates in homopolymer regions, potentially inflating IR calls due to mis-annotated intron-exon boundaries. Biologically, our stage-specific sampling (*Ac*3d, *Ac*10d, *Ac*21d) captures labor-division transitions where AF-mediated promoter switching regulates rapid transcriptional reprogramming (e.g., *Vg6* and *CYP15A1* in Figure 3). In contrast, studies using pooled age/tissue samples may emphasize constitutive processes such as IR. Such methodological distinctions highlight how AS landscapes are dynamically shaped by experimental design, yet collectively advance understanding of post-transcriptional complexity in honeybees. In summary, the proportion and types of AS events in genes differ across species, reflecting the species-specific characteristics to some extent. Moreover, the proportion of AS events within the genes of the same species differs among ages, highlighting the significant role of these events in growth and development.

APA regulates RNA function, localization, stability, and translational efficiency by producing transcripts with different coding sequences or 3′ UTR lengths, thereby increasing transcript diversity [29]. The phenomenon of APA has been confirmed in a wide range of species, such as *D. melanogaster* [30], *Mus musculus* [31], and *N. ceranae* [25]. This study is the first to characterize the number and types of APA sites in *A. cerana cerana* of different ages. At 3, 10, and 21 days, 1441, 1103, and 1328 genes with different numbers of APA sites were identified, respectively, with the highest number of genes containing two APA sites at 3 days (627, 43.51%), 10 days (564, 51.13%), and 21 days (548, 41.26%) and the fewest genes containing five APA sites at 3 days (115, 7.98%), 10 days (59, 5.35%), and 21 days (120, 9.04%). Chang et al. used PacBio SMRT sequencing technology to detect APA sites in cattle (*Bos taurus*), revealing that genes containing one APA site were most common (1929, 72.98%), and genes with five APA sites were least common (21, 0.79%) [32]. In sorghum (*Sorghum bicolor* L. Moench), genes containing one APA site were most common (i.e., 3289, accounting for 29.86% of genes [33]). In *A. mellifera*, three APA sites are most common in genes related to endocytosis [31]. In this study using PacBio SMRT sequencing technology, genes with two APA sites were most common in *A. cerana cerana* of different ages, and there were differences in the number of APA genes among age groups. However, whereas most genes contain a single APA site in many taxa, most genes in *A. cerana cerana* contained two APA sites. The mechanism needs further study. The motifs upstream of APA sites can have significant implications for gene expression and function. The presence of specific motifs can influence the usage of different polyadenylation sites (PAS). These motifs serve as binding sites for polyadenylation factors, which mediate the cleavage and addition of the poly(A) tail to pre-mRNAs. In our study, five motifs were identified upstream of the sites, namely AAUAAA, GUAC, UUUUUUUU, UGUAA, and AAUAAA. Further studies are needed to determine how these motifs regulate the formation of *A. cerana cerana* APA sites and control development or functional transformation. Using RT-PCR, we verified the results for four randomly selected genes associated with growth and development (*Vg6, Acsl1, esterase FE4,* and *PTPRR*) (Figure 7C), confirming the reliability of the alternative polyadenylation (APA) events identified in this study. Therefore, these results provide an addition to the reference genome annotation of *A. cerana cerana*.

The vitellogenin gene *Vg6* may play a significant role in the environmental adaptation, reproductive development, and functional transformation of worker bees in *A. cerana cerana*. The *vitellogenin* (*Vg*) gene encodes a multi-domain apolipoprotein, which is the precursor of vitellin. It provides nutritional reserves for embryonic development in oviparous and viviparous animals. The *Vg* gene family shows diversity in different species, and its encoded peptides are different in structure and function. *Vg* gene plays an important role in the differentiation of honeybees. Wang et al. found that the expression level of the vitellogenin gene was regulated by m6A methylation during larval development [34]. The *Vg* gene mRNA of worker bee larvae showed a unique high m6A modification. After the m6A methylation was inhibited by chemical methods, the *Vg* gene mRNA accumulation increased, which induced the characteristics of queen development [34]. Chen et al. found that the specific vitellogenin gene was highly expressed in the ensheathing glia of the queen bee, and RNAi interference with the expression of this gene in bee larvae could significantly inhibit its transformation to the queen bee [35]. The *Vg* gene not only plays a role in reproduction and differentiation but also is related to the regulation of bee behavior. Harwood et al. have proved that the expression level of the *Vg* gene is closely related to the foraging behavior of worker bees. Silencing the vitellogenin gene by RNAi technology can significantly change the foraging behavior of worker bees [36]. *Vg6* is a specific member of the vitellogenin gene family, which mainly plays a role in the reproductive and immune processes of insects. In the *Lasioderma serricorne*, the expression of the *Vg6* and its receptor (*vGr*) gene can regulate the female reproductive efficiency. RNAi interference with *Vg6* will damage ovarian development and significantly reduce the number of eggs laid and the hatching rate of eggs [37]. The *Vg6* gene is also involved in insect immune response. In the tobacco beetle, the protein encoded by the tobacco beetle contains immune-related domains. In some insects, it shows antibacterial and antiparasitic functions [38]. In *A. mellifera*, the queen bee will transfer the immune elicitor to the developing oocyte, which can realize the high specificity of larval immune initiation [34]. However, research on the role of the *Vg6* gene in the physiological regulation of *A. cerana cerana* remains unclear. In this study, *Vg6* was expressed in worker bees of different ages and underwent AS and APA. Additionally, our study found that the log2FC values of Vg6 in *Ac*3d, *Ac*10d, and *Ac*21d worker bees were 0.056, 0.971, and −1.027, respectively, and were significantly higher in *Ac*10d than in *Ac*3d and *Ac*21d. Furthermore, when comparing AS events among the three age groups, *Ac*10d had the lowest proportion of ASGs, at 42.57%, consistent with our previous research results showing that *Ac*10d had the fewest genes associated with GO functions (i.e., 3524) [39]. Therefore, we hypothesize that *Vg6*, which regulates growth and development through various mechanisms, is a key gene in the reproductive development and functional transformation of worker bees in *A. cerana cerana*. Ten days is a critical age for the development and functional transformation of worker bees in *A. cerana cerana*. Prior to this time point, worker bees can develop the ability to lay unfertilized eggs under temporary queenless conditions, while worker bees over ten days of age primarily shift their focus to foraging activities, such as pollen and nectar collection. This transition is accompanied by significant changes in task allocation and a more defined division of labor. Phenoloxidase 2 (*PPO2*) may regulate the growth, development, and immune mechanisms of *A. cerana cerana* at different developmental stages. Phenoloxidase activity is involved in regulating both development and immune defense in insects, such as *Aedes aegypti* [40], *Pieris rapae* (L.) [41], and *Bactrocera tau* [42]. PPO is an immune defense barrier in insects against the invasion of foreign substances, and it is transformed into activated phenoloxidase (PO) through a cascade reaction; it regulates physiological processes, such as melanization and sclerotization [43,44]. In *D. melanogaster*, PPO1 and *PPO2* in the hemolymph are activated in response to pathogen infections, enhancing immunity by regulating melanization [45]. Jiwon shim found that the crystal cells of *Drosophila melanogaster* under oxygen stress can capture and transport oxygen through PPO2 protein, thus helping *Drosophila* larvae maintain oxygen balance in different oxygen environments [46]. At present, there is no report on the physiological regulation of *PPO2* in *A. cerana cerana*. In this study, the log2FC values for *PPO2* were 1.143, −0.433, and −0.711 in *Ac*3d, *Ac*10d, and *Ac*21d, respectively. Therefore, the expression of *PPO2* is closely related to the growth and development of *A. cerana cerana* and varies among developmental stages. Worker bees rest or clean the hive at one to three days old, mainly feed larvae and the queen at four to 12 days old, and mainly engage in foraging outside the nest at 21 days old [47,48,49]. Thus, it is speculated that worker bees at 3 days old mainly work inside the hive, and their immune mechanisms are not yet fully developed, mainly regulating the immune mechanism through PPO. Worker bees gradually develop a complete immune system with increasing age, and immunity at 10 and 21 days old is regulated by various mechanisms to cope with the complex external environment.

In this study, we re-annotated the full-length transcriptome of *A. cerana cerana* using PacBio third-generation sequencing technology and identified a large number of isoforms and APA genes in three samples at different stages, providing a basis for studying the role of post-transcriptional regulation in the functional transformation of honeybees. There were differences in the proportion and type of AS events among developmental stages, indicating that AS plays an important role in growth and development. APA regulates the function and stability of RNA by producing transcripts of different lengths, increasing transcript diversity. This study identified and verified APA loci in *A. cerana cerana* at different ages for the first time. The number of APA genes differed with respect to age, and most genes contained two APA loci, which may be related to the development or functional transformation. The results of this study provide a valuable supplement to the annotation of the reference genome of *A. cerana cerana*. The egg reduction protein gene *Vg6* plays a key role in the reproductive development and functional transformation of *A. cerana cerana* workers, while *PPO2* plays a role in regulating growth, development, and immunity at different developmental stages. These findings not only reveal the complexity of post-transcriptional regulation in honeybees but also offer new insights into their growth and immune responses. a novel perspective for understanding the growth and immune responses of honeybees. However, our current study has certain limitations that warrant consideration. First, while we observed age-associated patterns in AS and APA events, we cannot fully disentangle the confounding effects of age and task-specific demands on these molecular processes due to the inherent behavioral and physiological transitions in honeybee development. Second, although the data highlight correlations between splicing/polyadenylation events and developmental stages, the mechanistic links connecting these post-transcriptional modifications to specific biological outcomes—particularly their precise roles in regulating reproductive capacity, immune function, and transcriptome plasticity—remain underexplored. Future studies integrating functional validation (e.g., isoform-specific knockdown) and behavioral experiments to clarify the causal relationships underlying these regulatory networks.

## 4. Materials and Methods

### 4.1. Honey Bees

Three colonies at the Guizhou Institute of Modern Agricultural Development (E106°66′, N26°49′) were selected as the source of *A. cerana cerana* workers. Samples were collected in April 2022. The newly emerged bees (*Ac*0d) were marked with color and returned to the hive, where they were later recaptured at the appropriate age. The 3-(*Ac*3d) and 10-(*Ac*10d) workers were captured in the hive. The 21-(*Ac*21d) workers were captured at the nest entrance. For Illumina RNA-seq sequencing, 10 honeybees were used per sample with 3 biological replicates. For PacBio Sequel sequencing, 15 honeybees were used per sample. Total RNA was obtained using a TRizol Re-gent kit (Thermo Fisher, Waltham, MA, USA).

### 4.2. Full-Length Transcriptome Data Source

The samples of *Ac*3d, *Ac*10d, and *Ac*21d groups of the *A. cerana cerana* workers were sequenced by PacBio technology, yielding high-quality full-length transcriptome data. Raw PacBio reads were processed through SMRTlink (v13.1) to generate consensus sequences, which were subsequently error-corrected against Illumina-derived high-quality data. The polished sequences were aligned to the reference genome (gca_029169275.1) via GMAP (v2011-10-16) [50]. Raw reads were 537,698, 329,469, and 418,947 bp; median lengths (N50) were 201,799, 197,011, and 217,070 bp; and average read lengths were 132,366, 129,800, and 140,398 bp, respectively. The hierarchical n*log(n) algorithm was used to cluster the FLNC sequences of the same transcript to remove redundancy, leading to the identification of 25,041, 14,738, and 20,963 non-redundant full-length transcripts. The N50 values were 3523, 3064, and 3189 bp, with average lengths of 3134, 2712, and 2808 bp. The maximum read lengths were 11,425, 10,848, and 11,424 bp. The full-length transcriptome data provide a reliable basis for analyses of AS and APA in *A. cerana cerana*.

### 4.3. Alternative Splicing Isoform Analysis

Based on a comparison between the consensus sequence and reference genome, TAPIS software (TAPIS version 1.2.1) was used to further correct the consensus sequence, cluster, and remove redundancy to obtain the final high-quality isoforms. SUPPA software (SUPPA version 2.4) [51] was used to identify the AS types of genes using default parameters. The types of AS mainly include skipped exon (SE), mutually exclusive exon (MX), alternative 5′ splice site (A5), alternative 3′ splice site (A3), alternative first exon (AF), alternative last exon (AL), and retained intron (RI). Use the Benjamini Hochberg program to control FDR. Predicted AS event types were counted and visualized in a pie chart using R software (R version 4.2.3). Fold change ≥ 2 (fold change was calculated based on TPM (transcripts per million)) and FDR < 0.05 were used as criteria for screening differential AS events (DASEs) from *Ac*3d vs. *Ac*10d, *Ac*3d vs. *Ac*21d, and *Ac*10d vs. *Ac*21d.

### 4.4. Bee Poly(A) Analysis

The APA loci of genes were identified using the TAPIS pipeline. MEME software (MEME software version 5.3.3) was used to analyze the region 50 bp upstream of the poly(A) splice site of each transcript to identify motifs. The parameters were as follows: -norc, -meme minw 6, -meme maxw 6, -spamo skip, -fimo skip. The poly(A) loci were generated by PacBio deep sequencing, and the differential poly(A) loci were quantified. Fold change ≥ 2 and FDR < 0.05 were used as criteria to screen genes with evidence for differential alternative polyadenylation genes (DAPAGs) from *Ac*3d vs. *Ac*10d, *Ac*3d vs. *Ac*21d, and *Ac*10d vs. *Ac*21d.

### 4.5. GO Enrichment and KEGG Annotation Analysis

The categorization of isoforms according to Gene Ontology (GO) was performed using GOseq software (GOseq version 1.60.0) [52]. For pathway analysis, the Blastall tool was employed to compare isoforms against the KEGG (Kyoto Encyclopedia of Genes and Genomes) database (https://www.kegg.jp/) (Accessed on 7 February 2023). Perform pathway enrichment analysis using KOBAS (3.0) [53]. Subsequently, the results were visualized using relevant tools from the OmicShare platform (https://www.omicshare.com/tools/) (Accessed on 26 June 2025).

### 4.6. RNA Extraction and RT-PCR

Total RNA was extracted using the RNA Extraction Kit (TaKaRa, Dalian, China), and its integrity was checked using a Bioanalyzer (Thermo Fisher, Waltham, MA, USA). The PrimeScript™ RT Reagent Kit (TAKARA) was used to reverse transcribe 200 ng of RNA into cDNA. For PCR validation of AS isoforms, 1 µL of cDNA was used in a reaction volume of 20 µL using TransStart FastPfu DNA Polymerase (TaKaRa, Dalian, China). Gene-specific primers were designed to span the predicted splice sites using Primer Premier5 software. PCR conditions were 2 min at 95 °C, followed by 35 cycles of 95 °C for 20 s, 45 °C for 20 s, and 72 °C for 60 s, with a final extension at 72 °C for 1 min. Samples were separated on a 2% agarose gel. For APA validation, we used 3′ RACE (TaKaRa, Dalian, China) for cDNA synthesis and PCR amplification. cDNA was prepared using adaptor primers, and PCR was carried out using 3′ RACE outer primers and gene-specific forward primers (Appendix A). Inner primer PCRs were carried out with the products obtained using outer primers. Samples were separated on a 2.0% agarose gel.

## 5. Conclusions

In summary, the study employed PacBio sequencing to re-annotate the *A. cerana cerana* transcriptome, identifying isoforms and APA genes across different developmental stages, thereby providing valuable insights into post-transcriptional regulation in honeybee development. The results underscored the critical role of AS in developmental regulation and the contribution of APA to transcriptome complexity. Notably, the study uniquely identified APA loci, with most genes containing two sites, potentially linked to developmental processes in honeybees. Furthermore, the results demonstrate that *Vg6* and *PPO2* critically regulate honeybee reproduction and immunity, suggesting their roles in colony health and evolutionary adaptation.

## Figures and Tables

**Figure 1 ijms-26-07859-f001:**
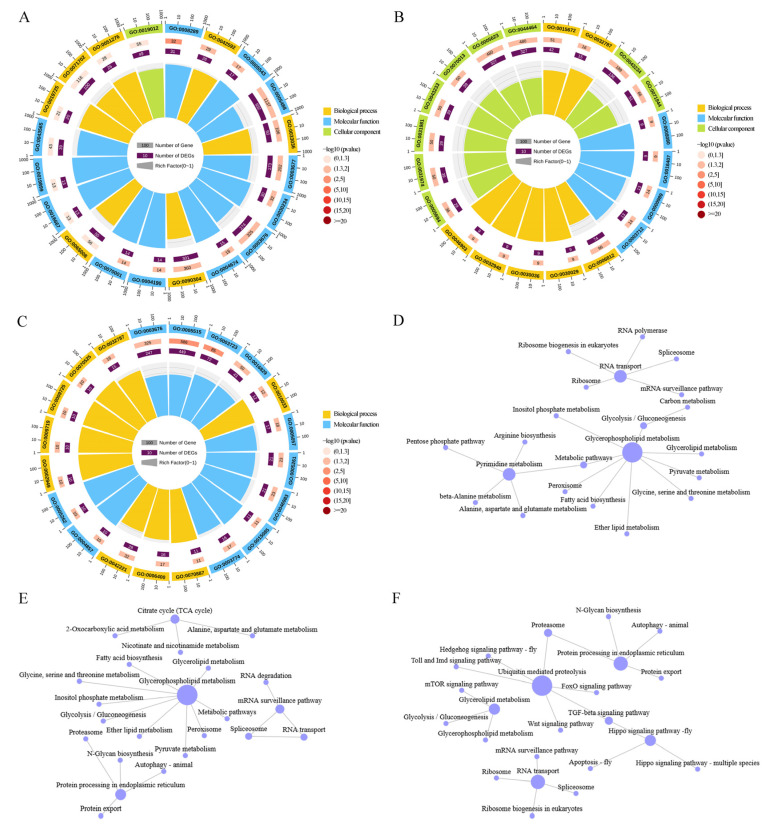
GO and KEGG database annotation of AS genes. (**A**–**C**) GO analysis of the AS genes revealed significant enrichment for different functions in *Ac*3d, *Ac*10d, and *Ac*21d. (**D**–**F**) KEGG analysis showed that the AS genes were involved in different pathways in *Ac*3d, *Ac*10d, and *Ac*21d.

**Figure 2 ijms-26-07859-f002:**
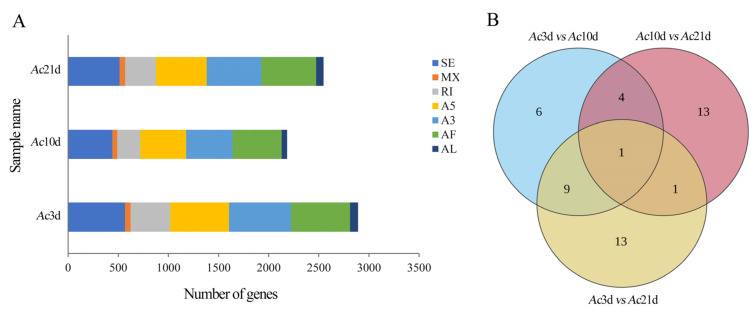
Types of AS events and a heat map of differential AS events (DASEs). (**A**) Number and proportion of seven AS types detected in *Ac*3d, *Ac*10d, and *Ac*21d. (**B**) Venn analysis of DASEs.

**Figure 3 ijms-26-07859-f003:**
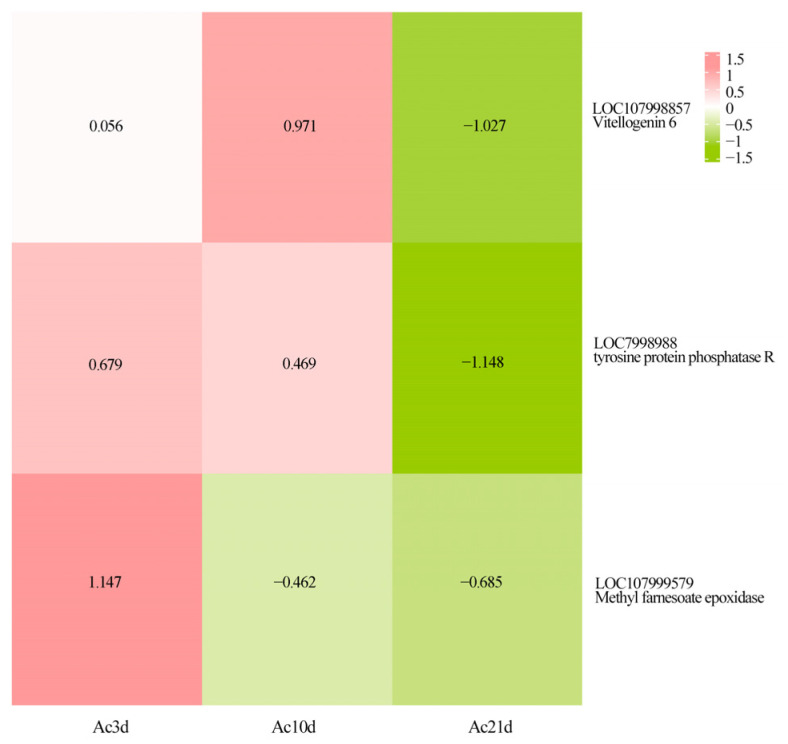
The heatmap of growth and development factor-associated DASEs.

**Figure 4 ijms-26-07859-f004:**
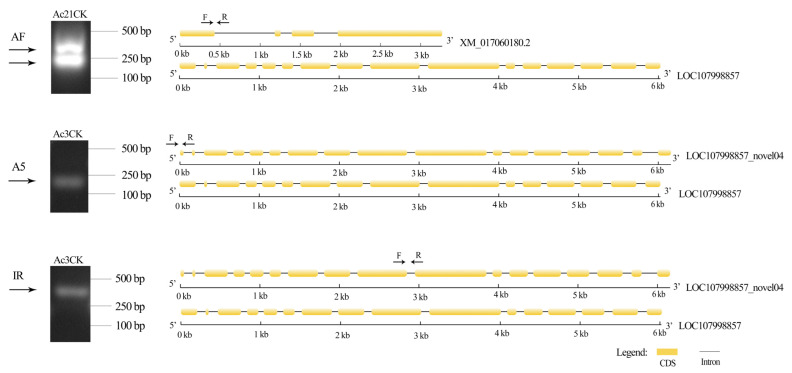
RT-PCR verification of DASEs. Two DASEs associated with growth and development were randomly selected for verification using RT-PCR. The results matched the predicted sizes from sequencing data.

**Figure 5 ijms-26-07859-f005:**
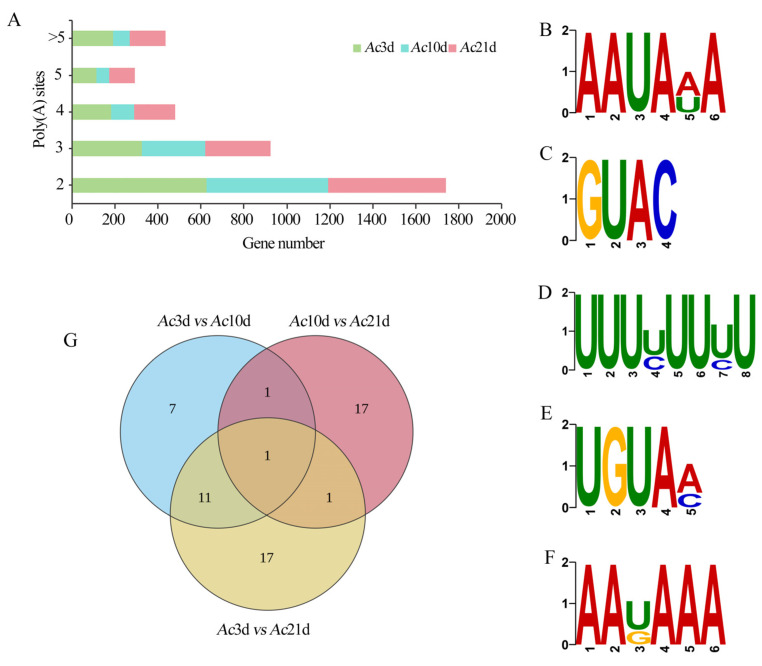
APA sites. (**A**) Distribution of APA sites in *Ac*3d, *Ac*10d, and *Ac*21d. (**B**–**F**) Five motifs, AAUAAA (**B**), GUAC (**C**), UUUUUUUU (**D**), UGUAA (**E**), and AAUAAA (**F**), were identified in the regions upstream of APA sites in *Ac*3d, *Ac*10d, and *Ac*21d. (**G**) Venn analysis of DAPAGs.

**Figure 6 ijms-26-07859-f006:**
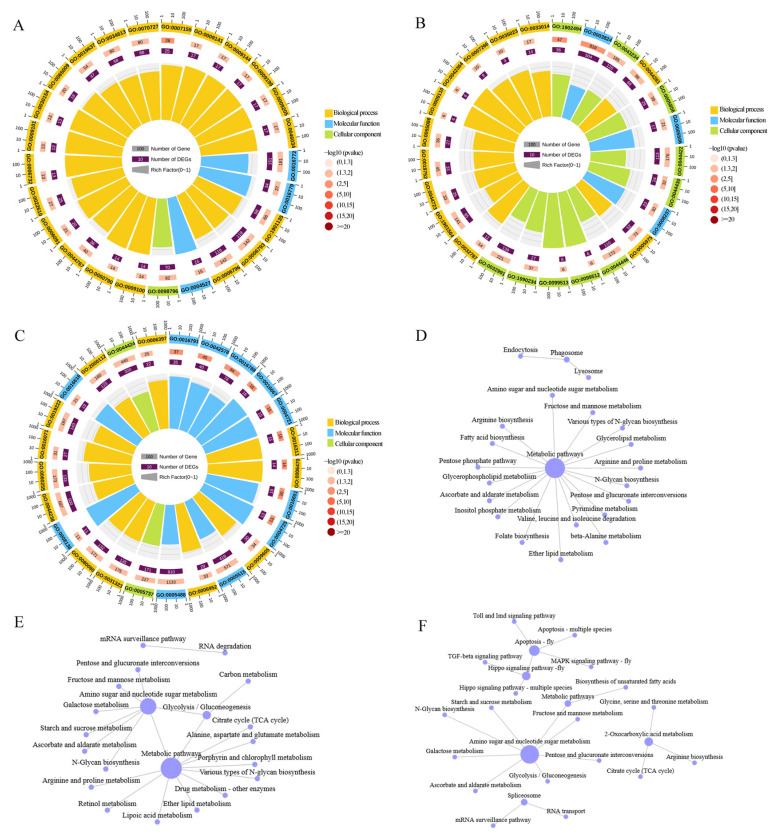
GO and KEGG database annotation of APA genes. (**A**–**C**) GO analysis shows that the APA genes in *Ac*3d, *Ac*10d, and *Ac*21d were significantly enriched in different terms. (**D**–**F**) KEGG analysis shows that the APA genes were significantly enriched in different pathways.

**Figure 7 ijms-26-07859-f007:**
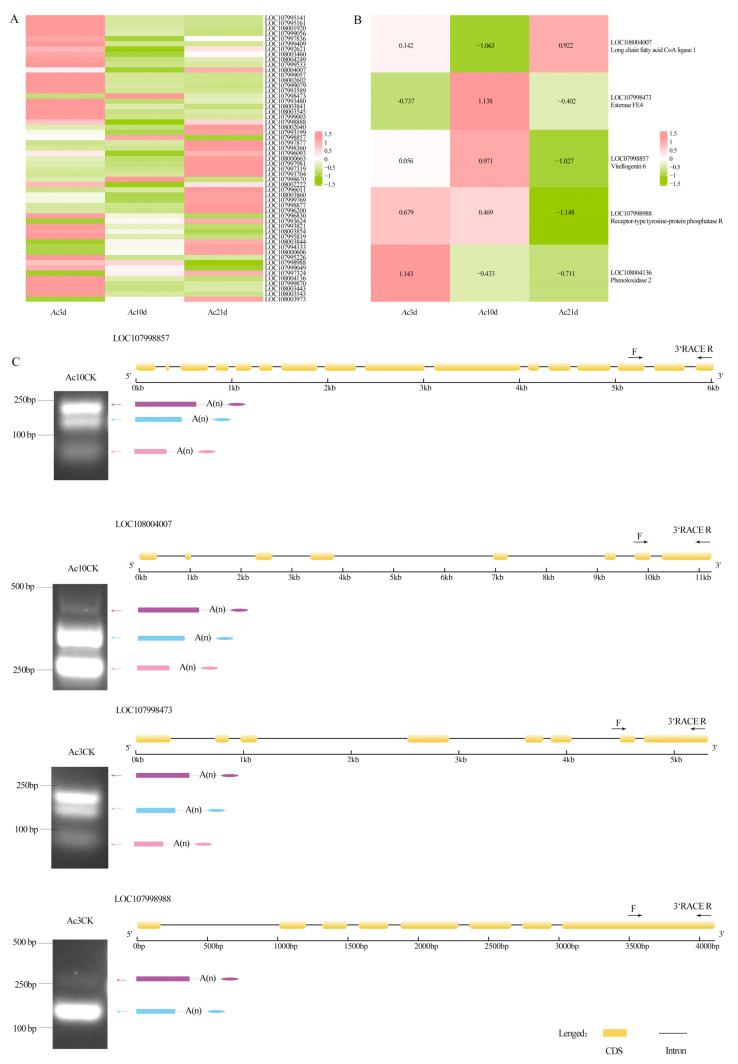
Analysis of DAPAGs. (**A**) Setting fold change ≥ 2 and FDR < 0.01 as thresholds, 54 significant DAPAGs were screened in *Ac*3d vs. *Ac*10d, *Ac*3d vs. *Ac*21d, and *Ac*10d vs. *Ac*21d. (**B**) The heatmap of growth and development factor-associated DAPAs. (**C**) Four DAPAGs associated with growth and development were randomly selected for verification by 3′ RACE. The sizes matched the predicted sizes from sequencing data.

## Data Availability

Sequence data that support the findings of this study have been deposited in the National Center for Biotechnology Information with the BioProject accession code SUB13867041.

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
