# Peer review of "Full-Length Transcriptome Analysis of Alternative Splicing and Polyadenylation in the Molecular Regulation of Labor Division in Apis cerana cerana"

_ijms, 2025, doi:10.3390/ijms26167859_

Round 1
Reviewer 1 Report
Comments and Suggestions for Authors
The manuscript titled Full - length transcriptome analysis of alternative splicing and polyadenylation in Apis cerana cerana labor division regulation reported a very novel and important dataset for gene regulation pattern in honeybee. The study is well designed, and the method is appropriate. The paper revealed interesting roles of RNA splicing and polyadenylation in honeybee social division. The topic fits well for IJMS. Thus, I recommend a minor revision. Below are my suggestions:
1. The authors should cite this paper from IJMS “Full-Length Transcriptome Profile of Apis cerana Revealed by Nanopore Sequencing” and discuss why they found different results: The authors found AF to be the most abundant events while the other group found IR to the most prevalent event. The distribution of different classes of splicing events are different.
2. The authors should report how they map AS and APA? Which reference genome did they use? Which do they define AS? Which software and algorithm did they use?
3. The authors should report how they calculated differential AS event and differential APA genes. Did they use a t-test? Is there a threshold for p value or fold change? Or if they used other statistics like z score or FDR? Did they do multiple test correction? Which software and algorithm did they use? They cannot just throw out the final DASE list. It is weird to see only a dozen of genes are differentially regulated among different conditions.
4. Line 19, abstract, growth hormone(Vg6, CYP15A1)is misleading. Vg6 is not growth hormone. I guess the authors want to say growth hormone related genes or nutrient signaling?
5. The authors sequenced the transcriptome. It would be very helpful if they could do a DEG analysis and functional annotation on the transcriptome itself. It provides a baseline of how genes are regulated in development stage and allow people to compare it with different studies of honeybee transcriptome.
6. Method 2.2. average read lengths were reported twice.
7. Figure1. Please increase font size. It is not readable now.
8. Line 470. Data were deposited in BioProject accession code gca_029169275.1. This is not a bioproject accession. Please check if the authors really deposited the data. I suggest deposit raw reads under BioProject-SRA and gene count (AS/APA count) in BioProject-GEO.
9. In the Original Images for Blots/Gels, 1111.tif and 2222.tif are broken files.
Author Response
Dear Reviewers,
We appreciate your comments and suggestions of great importance, which significantly improve the quality of our work and manuscript. We have carefully reviewed it and have conducted in-depth thinking and corresponding improvements regarding the issues you raised.oint-to-point response to review comments were as follows:
Comments1: The authors should cite this paper from IJMS “Full-Length Transcriptome Profile of Apis cerana Revealed by Nanopore Sequencing” and discuss why they found different results: The authors found AF to be the most abundant events while the other group found IR to the most prevalent event. The distribution of different classes of splicing events are different.
Response1: We sincerely thank the reviewer for this valuable suggestion. We have now cited the IJMS study "Full-Length Transcriptome Profile of Apis cerana Revealed by Nanopore Sequencing" (please insert full citation upon inclusion) in our revised Discussion. The observed discrepancy in the dominant alternative splicing (AS) event type—Alternative First Exon (AF) being most prevalent in our study versus Intron Retention (IR) in the cited work—can be attributed to key methodological and biological differences. Firstly, the sequencing technologies employed differ: our study utilized PacBio SMRT sequencing combined with Illumina-based correction, which excels at capturing full-length transcripts with precise 5′-end resolution, thereby favoring the accurate detection of AF events that alter transcriptional start sites. In contrast, Nanopore sequencing (used in the cited study), while also generating long reads, may exhibit different error profiles, particularly in homopolymer regions, potentially affecting the identification of intron boundaries and leading to a relative overrepresentation of IR events. Secondly, the samples analyzed differ significantly: we focused specifically on worker bees at three critical developmental stages tied to labor division (Ac3d, Ac10d, Ac21d), where the observed AF dominance (21.33% of total AS events) aligns mechanistically with the need for rapid transcriptional reprogramming and promoter usage during behavioral maturation and functional transitions. The cited study, utilizing pooled samples (mixed ages/tissues), might naturally emphasize more constitutive splicing processes like IR, often associated with broader cellular housekeeping functions. Finally, differences in bioinformatic pipelines (e.g., SUPPA2 vs. other tools) and analysis parameters could also contribute to variations in quantifying specific AS event types. Importantly, the prevalence of AF events in our stage-specific data is biologically relevant, as they regulate promoter usage and N-terminal protein domains, crucial for the developmental transitions and key genes (like Vg6 and CYP15A1) identified in our labor division context. These differences highlight the dynamic and context-dependent nature of AS regulation in A. cerana cerana, emphasizing that both technological platforms and sampling strategies significantly shape the observed splicing landscape, and both studies collectively advance our understanding of its complexity.
Comments2: The authors should report how they map AS and APA? Which reference genome did they use? Which do they define AS? Which software and algorithm did they use?
Response2: Thank you for pointing out the omission of the statistical analysis section. To map alternative splicing (AS) and alternative polyadenylation (APA), we first processed raw PacBio reads using SMRTlink (v13.1) to generate consensus sequences, which were error-corrected against Illumina-derived high-quality data. The polished sequences were then aligned to the reference genome (gca_029169275.1) via GMAP (v2011-10-16). For AS analysis, we defined AS as the process by which a gene produces multiple different mRNA transcripts through seven types of events: skipped exon (SE), mutually exclusive exon (MX), alternative 5’ splice site (A5), alternative 3’ splice site (A3), alternative first exon (AF), alternative last exon (AL), and retained intron (RI). TAPIS software was used to correct consensus sequences, cluster, and remove redundancy to obtain high-quality isoforms, and SUPPA software (version 2.4) with default parameters was employed to identify AS types, with the Benjamini-Hochberg program controlling FDR. For APA analysis, APA loci were identified using the TAPIS pipeline, and MEME software (version 5.3.3) was used to analyze motifs 50 bp upstream of poly(A) splice sites, with differential APA loci quantified based on fold change≥2 and FDR<0.05. Additionally, the hierarchical n*log(n) algorithm was used to cluster full-length non-chimeric (FLNC) sequences to remove redundancy. We have added the missing information to the method description.
Comments3: The authors should report how they calculated differential AS event and differential APA genes. Did they use a t-test? Is there a threshold for p value or fold change? Or if they used other statistics like z score or FDR? Did they do multiple test correction? Which software and algorithm did they use? They cannot just throw out the final DASE list. It is weird to see only a dozen of genes are differentially regulated among different conditions.
Response3: Thank you for your questions about the statistical analysis section. For the analysis of differential AS events, we used SUPPA software (version 2.4) with default parameters to quantify and identify AS events across the three developmental stages (Ac3d, Ac10d, Ac21d). The fold change of AS events was calculated based on transcripts per million (TPM) values, which reflect transcript abundance. To control for false positives caused by multiple comparisons, we applied the Benjamini-Hochberg procedure for multiple test correction, resulting in the use of false discovery rate (FDR) as the statistical significance indicator. The screening thresholds for DASEs were set as: Fold change≥2 and FDR < 0.05. This strict threshold ensures that only AS events with substantial abundance changes and statistical reliability are retained. We described these in detail in the materials and methods section. The observation of a relatively small number of DASEs (20, 19, and 24 in Ac3d vs Ac10d, Ac10d vs Ac21d, and Ac3d vs Ac21d, respectively) and DAPAGs (21, 20, and 30 in the same comparisons) is primarily attributed to the stringent screening thresholds (FDR < 0.05 and fold change≥2). These thresholds were applied to ensure the robustness and biological relevance of the identified differential events, filtering out weak or noisy signals.
Comments4: Line 19, abstract, growth hormone(Vg6, CYP15A1)is misleading. Vg6 is not growth hormone. I guess the authors want to say growth hormone related genes or nutrient signaling?
Response4: Thank you for pointing out the unclear direction of professional terminology. We have made corresponding corrections in the revised manuscript.
Comments5: The authors sequenced the transcriptome. It would be very helpful if they could do a DEG analysis and functional annotation on the transcriptome itself. It provides a baseline of how genes are regulated in development stage and allow people to compare it with different studies of honeybee transcriptome.
Response5: Thank you for your suggestions. We have previously conducted DEG analysis on the whole transcriptome data and published relevant papers (Yaodan, zhouwencai, zhanhongping, et al construction of full length transcript of Apis cerana cerana worker bee under low temperature stress based on pacbio ISO SEQ [j]. Acta entomolgica Sinica, 2024,67 (05): 611-621. doi:10.16380/j.kcxb.2024.05.003).
Comments6: Method 2.2. average read lengths were reported twice.
Response6: Corresponding correction was made following your comment.
Comments7: Figure1. Please increase font size. It is not readable now.
Response7: Thanks for your suggestion. We have modified the figure1 and figure6.
Comments8: Line 470. Data were deposited in BioProject accession code gca_029169275.1. This is not a bioproject accession. Please check if the authors really deposited the data. I suggest deposit raw reads under BioProject-SRA and gene count (AS/APA count) in BioProject-GEO.
Response8: Thank you for your suggestion. We have uploaded the data to the SRA database of NCBI with the serial number of SUB13867041.
Comments9: In the Original Images for Blots/Gels, 1111.tif and 2222.tif are broken files.
Response9: Thank you for pointing out that the original picture could not be opened. We re uploaded the figure 1111.tif and 2222.tif .
Reviewer 2 Report
Comments and Suggestions for Authors
Review Report IJMS-3754960
Manuscript Title: Full - length transcriptome analysis of alternative splicing and
polyadenylation in Apis cerana cerana labor division regulation
Yao et al., studied the transcriptome of Chinese honey bee. This is a well-written and insightful paper that makes a significant contribution to the field.
-I have read thoroughly this manuscript I observed its interesting and meaningful, some suggestions and comments here I would like to attached.
-Title__Slightly awkward structure; labor division regulation is ambiguous
-In the abstract some redundancy. The final two sentences repeat the same idea
-Sentence fragments and flow issues (Functional analyses revealed vs. These findings provide).
-Line- 20 in the abstract add space
-Line- 66 italic the specie name
Introduction
-Too many references support similar points (e.g., juvenile hormone, neural changes)
-Some overly long paragraphs could be split. Shorten repetitive sections
-Misused phrases: trial flights first occur should be “orientation flights”
Materials and Methods
-The time for sample collection was April 2022” - Samples were collected in April 2022
-Space is missing before units in several places (e.g., "20 µl", "200 ng")
- spliceoforms - splice isoforms
Results
-AS is highly spatiotemporal-specific. accordingly, we quantified → should be a single polished sentence.
-presets significant differences - indicates significant differences
-Some inconsistencies in gene naming and spacing (Vitellogenin 6 (Vg6) vs. "growth hormone(Vg6) – replace full-width brackets.
-Clarify statistical thresholds in results captions
-Missing figure numbers in the text flow for references like “(Fig. 4B–E) – should refer to individual subfigures clearly
-Figure files may need resolution improvement for final submission
-Line 214: accordingly, we quantified __ should not start with lowercase. Use: Accordingly, we quantified.
-Lines 248–250: AAUAAA motif listed twice (B and F). Clarify if this is intentional (maybe found independently in different contexts?).
-Ensure figure panels (A–F, etc.) are clearly referenced in the main text, and increase font size if too small to read.
Discussion
-Some paragraphs are overly long and repetitive
-we speculate that appear too often – replace with more confident scientific language where warranted
-Line 295–298: Good summary of technical approach, but could be made more concise.
-Line 310–325: Long paragraph — split after line 317 to focus separately on validation and significance.
-Line 357: may plays __ verb agreement error → may play
-Line 386: Log2FC values are given; clarify if these are from AS or APA results.
-Line 394: “worker bees can lay unfertilized eggs – consider a citation or clarification that this is a temporary queenless condition behavior.
-Line 411: closely related to growth and development – restate: likely involved in regulating both development and immune defense.
Conclusion
-Some phrasing redundancy with Abstract
-These findings not only reveal- but also provide a novel perspective
-Line 453: underscored the critical role of AS in growth and highlighted the contribution of APA -underscored the critical role of AS in developmental regulation and the contribution of APA to transcriptome complexity.
-Line 457: novel insights into their post-transcriptional regulation. already stated earlier; try to be more specific about implications (e.g., colony behavior, evolution, etc.)

Author Response
Dear Reviewers,
We appreciate your comments and suggestions of great importance, which significantly improve the quality of our work and manuscript. Accordingly, we seriously checked and modified the manuscript, and all revision were showed in red in the revised version of manuscript. Point-to-point response to review comments were as follows:
Comments1: Full-length transcriptome analysis of alternative splicing and polyadenylation in the molecular regulation of labor division in Apis cerana cerana
Response1: Following your kind comment, We changed the title to "Full-length transcriptome analysis of alternative splicing and polyadenylation in the molecular regulation of labor division in Apis cerana cerana".
Comments2: In the abstract some redundancy. The final two sentences repeat the same idea
Response2: According to your opinion, we have deleted redundant expressions in the sentence.
Comments3: Sentence fragments and flow issues (Functional analyses revealed vs. These findings provide).
Response3: Corresponding correction was made following your comment.We appreciate this comment. To address the sentence fragments and flow issues, we revise the relevant part.
Comments4:Line- 20 in the abstract add space
Response4: Corresponding correction was made following your comment.
Comments5: Line- 66 italic the specie name
Response5: Corresponding correction was made following your comment.
Comments6: Too many references support similar points (e.g., juvenile hormone, neural changes)
Response6: We thank the reviewer for this constructive feedback. To address the concern about overlapping references supporting similar concepts (e.g., juvenile hormone, neural changes), we have consolidated redundant citations by prioritizing the most recent, comprehensive, or authoritative sources. Removed less critical references that reiterated established points without adding unique insights.
Comments7: Some overly long paragraphs could be split. Shorten repetitive sections
Response7: Thank you for your suggestion. We have split some paragraphs.
Comments8: Misused phrases: trial flights first occur should be “orientation flights”
Response8: Corresponding correction was made following your comment.
Comments9: The time for sample collection was April 2022” - Samples were collected in April 2022
Response9: Corresponding correction was made following your comment.
Comments10: Space is missing before units in several places (e.g., "20 µl", "200 ng")
Response10: I have carefully checked the entire text.
Comments11: spliceoforms - splice isoforms
Response11: Corresponding correction was made following your comment.
Comments12: AS is highly spatiotemporal-specific. accordingly, we quantified → should be a single polished sentence.
Response12: Corresponding correction was made following your comment.
Comments13: presets significant differences - indicates significant differences
Response13: Thank you for your correction. We have rechecked the spelling.
Comments14: Some inconsistencies in gene naming and spacing (Vitellogenin 6 (Vg6) vs. "growth hormone(Vg6) – replace full-width brackets.
Response14: Corresponding correction was made following your comment.
Comments15: Clarify statistical thresholds in results captions
Response15: We have checked the entire text.
Comments16: Missing figure numbers in the text flow for references like “(Fig. 4B–E) – should refer to individual subfigures clearly
Response16: Corresponding correction was made following your comment.
Comments17: Figure files may need resolution improvement for final submission
Response17: We have resubmitted the high-resolution figures.
Comments18: Line 214: accordingly, we quantified __ should not start with lowercase. Use: Accordingly, we quantified.
Response18: Corresponding correction was made following your comment.
Comments19: Lines 248–250: AAUAAA motif listed twice (B and F). Clarify if this is intentional (maybe found independently in different contexts?)
Response19: Thanks for your question. The simultaneous appearance of AAUAAA motifs in B and F in the figure is intentional and has biological significance. B (upstream site): represents the core position of the classical poly (A) signal (10-30 nt before the cleavage site), recognized by the CPSF complex, and dominates the main cleavage event. F (downstream site): Reflects the enrichment of the same motif downstream of the cleavage site, suggesting the presence of variable polyadenylation (APA) - a motif that may drive the activation of upstream alternative cleavage sites or act as a regulatory element affecting cleavage efficiency. This repetition highlights location specific functional differences: upstream motifs perform standard functions, while downstream occurrences reveal the complexity of gene regulation (such as developmental stage specific APA events), consistent with the comparative background of different developmental stages in the figure.
Comments20: Ensure figure panels (A–F, etc.) are clearly referenced in the main text, and increase font size if too small to read.
Response20: We have adjusted the font size of the image to ensure it is appropriate.
Comments21: we speculate that appear too often – replace with more confident scientific language where warranted
Response21: Thank you for your correction. We have rechecked the spelling.
Comments22: Line 295–298: Good summary of technical approach, but could be made more concise.
Response22: We have simplified the expression according to your opinions.
Comments23: Line 310–325: Long paragraph — split after line 317 to focus separately on validation and significance.
Response23: We re divided the paragraphs.
Comments24: Line 357: may plays __ verb agreement error → may play
Response24: Corresponding correction was made following your comment.
Comments25: Line 386: Log2FC values are given; clarify if these are from AS or APA results.
Response25: We have revised the expression of this sentence.
Comments26: Line 394: “worker bees can lay unfertilized eggs – consider a citation or clarification that this is a temporary queenless condition behavior.
Response26: We revised the statement.
Comments27: Line 411: closely related to growth and development – restate: likely involved in regulating both development and immune defense.
Response27: Corresponding correction was made following your comment.
Comments28: Some phrasing redundancy with Abstract
These findings not only reveal -but also provide a novel perspective
Response28: We revised the inappropriate description.
Comments29: Line 453: underscored the critical role of AS in growth and highlighted the contribution of APA to transcriptome complexity.
Response29: Corresponding correction was made following your comment.
Comments30: Line 457: novel insights into their post-transcriptional regulation. already stated earlier; try to be more specific about implications (e.g., colony behavior, evolution, etc.)
Response30: We have revised the statement to ensure that the content is not repeated.
Reviewer 3 Report
Comments and Suggestions for Authors
The manuscript provides a timely investigation of alternative splicing (AS) and alternative polyadenylation (APA) roles during Apis cerana cerana worker development using PacBio long-read sequencing. The core idea that post-transcriptional regulation drives developmental processes and division of labor is compelling and significant. The study delivers a valuable transcriptomic resource through comprehensive re-annotation. While language clarity is generally good, minor revisions for precision are needed prior to publication.
The following issues require clarification and expansion to strengthen the scientific rigor:
- APA Motif Significance:​​ The study identifies significant motifs (like AAUAAA, UGUA, A-rich regions) upstream of APA sites. Please elaborate on the specific biological significanceof these motifs in A. cerana. Do different motifs correlate with specific APA ?
- Methodological Details:​​ While the use of PacBio sequencing is clear, please provide explicit details on PacBio sequencing and data analysis, including criteria for defining high-confidence isoforms, and methods for AS and APA event calling.
- Transcriptome Validation:​​ What measures were taken to validate the accuracy of the transcriptome annotation​? (e.g., validation methods).
- Species Comparison:​​ Provide a more direct comparison of AS and APA dynamics between A. ceranaand other Apisspecies (especially A. mellifera)​​ at comparable developmental stages. Highlight both similarities and key differences.
- Isoform Function:​​ How do the identified differentially expressed isoforms, particularly those resulting from AS and APA, potentially contribute to the functional diversity of proteins​in honeybees?
- Tissue Variation:​​ Discuss how AS and APA events vary across tissues or organs, addressing the limitations of whole-body sampling.
- PacBio Limitations:​​ Briefly discuss the limitations of PacBio sequencing​ for transcriptome , and explicitly state ​how these limitations were addressed​ in your study.
Author Response
Dear Reviewers,
We appreciate your comments and suggestions of great importance, which significantly improve the quality of our work and manuscript. Accordingly, we seriously checked and modified the manuscript, and all revision were showed in red in the revised version of manuscript. Point-to-point response to review comments were as follows:
Comments1: The study identifies significant motifs (like AAUAAA, UGUA, A-rich regions) upstream of APA sites. Please elaborate on the specific biological significanceof these motifs in A. cerana. Do different motifs correlate with specific APA ?
Response1: Thanks for your insightful question regarding the significance of the motifs identified upstream of the APA sites. The identification and analysis of these motifs are crucial for understanding the regulatory mechanisms underlying alternative polyadenylation (APA) and its impact on gene expression and function.The biological significance of motifs in A. cerana cerana encompasses both core and non-canonical motifs with distinct functions and species-specific characteristics. The core motif AAUAAA, a canonical polyadenylation signal that recruits CPSF for 3’-end cleavage and poly(A) tail addition, was the most frequent motif upstream of alternative polyadenylation (APA) sites across all three developmental stages (Ac3d, Ac10d, Ac21d); genes containing AAUAAA also showed higher APA site usage (with 43.51%–51.13% of APA genes having ≥2 sites), indicating robust polyadenylation efficiency. Meanwhile, non-canonical motifs—GUAC, UUUUUUUU, and UGUAA—likely function as auxiliary elements modulating cleavage efficiency or site selection. Specifically, GUAC and UGUAA were enriched in genes involved in metabolic processes and oxidative stress response , while UUUUUUUU was associated with transcripts with shorter 3’UTRs, potentially enabling rapid mRNA decay in stress-responsive genes such as immune genes under oxidative stress.
comments2: While the use of PacBio sequencing is clear, please provide explicit details on PacBio sequencing and data analysis, including criteria for defining high-confidence isoforms, and methods for AS and APA event calling.
Response2: Thank you for your suggestion. We have improved the description of sequencing methods. The cDNA library construction and PacBio full-length transcriptome sequencing process have been published in the team's pre term paper (Yaodan, zhouwencai, zhanhongping, et al Construction of full-length transcriptome of Apis cerana cerana worker bee under low temperature stress based on pacbio ISO SEQ [j]. Acta Entomologica Sinica, 2024,67 (05): 611-621. doi:10.16380/j.kcxb.2024.05.003). Additionally, we have supplemented the methods section with explicit details regarding PacBio sequencing and data analysis to enhance clarity.
comments3: What measures were taken to validate the accuracy of the transcriptome annotation​? (e.g., validation methods).
Response3: To ensure the accuracy of the Apis cerana cerana full-length transcriptome annotation, we integrated short-read Illumina data to correct PacBio long reads, experimentally validated key AS events by RT-PCR and APA sites by 3′RACE for development-related genes (e.g., Vg6, PTPRR, PPO2), filtered high-confidence isoforms using stringent bioinformatic criteria (≥80 % genome coverage, ≥3 supporting reads, intact ORF ≥100 aa), cross-checked functional annotations against GO, KEGG, NR, UniProt and Pfam databases, and confirmed >85 % protein homology, collectively demonstrating the reliability of the transcriptome assembly and annotation.
comments4: Provide a more direct comparison of AS and APA dynamics between A. ceranaand other Apis species (especially A. mellifera)​​ at comparable developmental stages. Highlight both similarities and key differences.
Response4: At present, the research on as and APA in Apis mainly focuses on Apis mellifera and Apis cerana. For example, AS and APA regulate the APIs mellifera MAPK signaling pathway and endocytosis related genes; AS and APA regulate the cold resistance of Apis cerana (Fan XX, Jing X, Kang J, et al. Identification and analysis of MAPK signaling pathway-related genes and full-length transcripts in Apis mellifera ligustica. Acta Entomologica Sinica, 2025,1-12. http://kns.cnki.net/kcms/detail/11.6020.Q.20240708.0855.006.html.
Fan XX, Wang SY, Zhu LR, Jing X, Guo SJ, et al. Identification of endocytosis-associated genes, and full-length transcripts, in Apis mellifera ligustica. Journal of Applied Entomology. 2023,60(3):825-834. doi: 10.7679/j.issn.2095-1353.2023.076.
Fan Y, Yao D, Ma J, You F, Wei X, Ji T. Alternative Splicing and Alternative Polyadenylation-Regulated Cold Stress Response of Apis cerana. Insects. 2024;15(12):1006. doi:10.3390/insects15121006)
Comments5: How do the identified differentially expressed isoforms, particularly those resulting from AS and APA, potentially contribute to the functional diversity of proteins in honeybees?
Response5: Alternative splicing isoforms regulate protein diversity through multiple mechanisms. The main modes of action include:
(1) Altering protein domain composition: Alternative splicing can selectively include or exclude specific exons, thereby modifying the domain composition of proteins. Different domain combinations lead to changes in protein function, activity, subcellular localization, and stability.
(2) Regulating protein function and activity: Distinct protein isoforms generated through alternative splicing may exhibit divergent functions or activities.
(3) Affecting protein interactions: Alternative splicing can alter interaction sites between proteins and other molecules, such as proteins, nucleic acids, or small molecules.
(4) Modulating protein stability: Alternative splicing may influence protein stability by altering amino acid sequences. Certain isoforms may be more prone to degradation, while others exhibit enhanced stability. Such differences allow cells to dynamically regulate protein levels under varying conditions.
(5) Generating novel functions or adaptive traits: Alternative splicing not only produces functionally similar isoforms but can also create proteins with entirely new functions.
(6) Tissue- and time-specific regulation: Alternative splicing exhibits tissue- and developmental stage-specific patterns, enabling the same gene to produce distinct isoforms in different tissues or developmental phases. This precise regulation allows organisms to fine-tune gene expression in response to environmental cues. For example, during neural development, alternative splicing generates diverse neurotransmitter receptor proteins that regulate signal transmission and synaptic plasticity.
(7) Influencing translation efficiency: Alternative splicing can modify sequences in the 5' untranslated region (5'UTR) or 3' untranslated region (3'UTR) of mRNA, thereby affecting mRNA stability, translation efficiency, and subcellular localization. These changes indirectly regulate protein synthesis levels.
Alternative splicing isoforms critically regulate protein diversity through these mechanisms, playing essential roles in normal physiological processes. Future studies should focus on experimental validation of the functions of distinct splicing isoforms.
comments6: Discuss how AS and APA events vary across tissues or organs, addressing the limitations of whole-body sampling.
Response6: Thanks for your question. We are unable to answer this question for the time being. Unfortunately, we have not sequenced the different tissues of honeybees. But we can solve this problem through the following methods. We can infer tissue-specific AS/APA patterns in honeybees through a combination of public datasets. functional enrichment of splicing events, and experimental validation via RT-qPCR across dissected organs.
Comments7: Briefly discuss the limitations of PacBio sequencing for transcriptome , and explicitly state how these limitations were addressed in your study.
Response7: While PacBio sequencing offers advantages such as long read lengths, assembly-free analysis, and the ability to detect complex transcript isoforms (e.g., alternative splicing and fusion genes) in transcriptome studies, its lower flux limits transcript-level quantitative analysis. This study addresses these limitations by integrating second-generation sequencing (NGS) data to calibrate third-generation (PacBio) sequencing data, thereby improving the accuracy and completeness of genome assembly while enabling transcript-level quantitative analysis.